# Functional Feed with Bioactive Plant-Derived Compounds: Effects on Pig Performance, Muscle Fatty Acid Profile, and Meat Quality in Finishing Pigs

**DOI:** 10.3390/ani15040535

**Published:** 2025-02-13

**Authors:** Maria Chiara Di Meo, Ilva Licaj, Romualdo Varricchio, Mauro De Nisco, Romania Stilo, Mariapina Rocco, Anna Rita Bianchi, Livia D’Angelo, Paolo De Girolamo, Pasquale Vito, Armando Zarrelli, Ettore Varricchio

**Affiliations:** 1Department of Sciences and Technologies (DST), University of Sannio, 82100 Benevento, BN, Italy; mardimeo@unisannio.it (M.C.D.M.); illicaj@unisannio.it (I.L.); romstilo@unisannio.it (R.S.); rocco@unisannio.it (M.R.); vito@unisannio.it (P.V.); 2Department of Sciences, Roma Tre University, 00146 Roma, RM, Italy; romualdo.varricchio@uniroma3.it; 3Department of Sciences, University of Basilicata, 85100 Potenza, PZ, Italy; mauro.denisco@unibas.it; 4Department of Biology, University of Naples Federico II, 80126 Naples, NA, Italy; annarita.bianchi@unina.it; 5Department of Veterinary Medicine and Animal Production, University of Naples Federico II, 80137 Naples, NA, Italy; livia.dangelo@unina.it (L.D.); paolo.degirolamo@unina.it (P.D.G.); 6Department of Chemical Sciences, University of Naples Federico II, 80126 Naples, NA, Italy; zarrelli@unina.it

**Keywords:** *Olea europaea* L. polyphenols, fatty acids, functional feed, antioxidants, pig performances

## Abstract

Due to their potential use in animal nutrition, the phenolic compounds present in olive tree by-products and their beneficial effects on livestock are a topic of great scientific and industrial interest. Considering this, the present study aimed to investigate the effects of Sulla plant and *Olea europaea* L. leaf supplementation on pig performance and the fatty acid profile of pig meat. In this study, 30 commercial hybrid pigs were fed a diet enriched with *O. europaea* L. polyphenols (300 mg/day) during the finishing period. Our hypothesis was that the inclusion of olive leaves could be a useful strategy for (i) the use of natural rather than synthetic antioxidants in pig feed; (ii) not altering the growth performance of pigs; and (iii) positively influencing the composition of the meat fatty acid profile, where the MUFA/SFA ratio was increased and the n-6/n-3 ratio was reduced, maintaining the optimal range. The results show that the production of functional feeds with natural extracts from *O. europaea* L. for pigs can be a valid strategy to improve meat quality.

## 1. Introduction

In recent years, the principles shown in the European Commission’s action plan (2006) on reducing the use of antibiotics in animal breeding and considering ecological transition approaches, to transform “no-food” matrices into important resources and not waste, have stimulated scientific research into innovative solutions for a better balance between the environment, food systems, biodiversity, and circularity of resources [1]. Specifically, the use of agro-industrial coproducts in animal feed, from the olive oil sector, which holds significant economic and social importance in the Mediterranean area [2], offers a promising natural alternative. Due to their high polyphenol content, their co-products can enhance performance parameters, improve animal welfare, and preserve food quality, especially meat and meat products [3,4,5,6].

Olive oil co-products, such as olive leaves, are rich in polyphenols, natural compounds with high biological value. When included in the diets of monogastric animals (pigs), these compounds exhibit various beneficial effects, including anti-inflammatory, antioxidant, antibacterial, antiproliferative, and antifungal activities [7,8,9]. Among the bioactive molecules found in olive leaves, oleuropein and hydroxytyrosol are the most abundant [10,11] and notable for their beneficial properties. Oleuropein is known to localize within biomembranes and alongside hydroxytyrosol, acts as a scavenger of peroxyl radicals near the membrane surface, interfering with their chain propagation [12]. Paiva-Martins et al. [13] highlight that nutritional interventions involving the use of bioactive molecules extracted from plant matrices and olive supply chain co-products are crucial for enhancing the fatty acid profile, antioxidant content, and lipid stability of pig meat. In this context, feeding animals with natural antioxidants could be used as a vehicle for these compounds in the circulatory system, facilitating their distribution and retention in tissues, thereby positively influencing meat quality characteristics [14].

Incorporating a phenolic extract with high antioxidant content into the diet, in association with bioactive molecules from pasture grasses such as Sulla, represents a novel strategy in farm animals, to improve their growth performance, animal welfare, and production quality. This approach aligns with the goals of sustainable agriculture and animal breeding. According to the literature, consuming pasture grasses rich in antioxidants, especially vitamins, carotenoids, and flavonoids, can protect meat against oxidation, improving meat quality and animal welfare [15]. Specifically, Sulla (*H. coronarium* L.), a short-lived perennial legume native to the Mediterranean area and cultivated as a biennial forage for grazing and/or hay or silage production, has demonstrated positive effects on the productivity of several animal species [16,17,18].

The positive effects of this plant species can be attributed to its high protein content, substantial levels of secondary metabolites (phenolic compounds), favorable ratio of degradable to structural carbohydrates, presence of proanthocyanidins, and moderate content of condensed tannins (CTs). These attributes vary significantly according to species and growth stages [16,18,19]. The intake of CT is associated with the transfer of phenolic compounds to various tissues thus contributing to an increase in the antioxidant capacity and oxidative stability of animal products [20,21]. Pasture grasses like Sulla offer high functional value in the diet, boosting the overall beneficial activity [22].

Therefore, the presence of secondary metabolites in this plant species could offer valuable support to limit drug use and improve livestock health [15]. Given the lack of studies in pigs on the combined effects of Sulla intake and antioxidant compounds extracted from olive oil supply chain coproducts, this study aims to address this gap. The objective is to formulate a feed functionalized with molecules from *O. europaea* L. for pigs, combined with a diet incorporating pasture grasses, to enhance precision feeding. This perspective aligns with the European Union plans and programs, including the One Health holistic approach, the European Green Deal, and Farm to Fork strategies, which emphasize sustainable and integrated food systems.

In recent years, there has been a growing interest among consumers for not only the nutritional but also the functional properties of animal-derived foods destined for human consumption, particularly meat. Specifically, the fatty acid composition of meat influences several quality parameters, such as its nutritional and functional quality, shelf life, and rheological characteristics [13,18]. The composition and content of the fatty acids in pig meat have garnered attention due to their impacts on human health [23]. Recently, the presence of a high saturated fatty acid (SFA) content in meat has driven scientific research toward identifying and supplementing livestock diets with natural bioactive molecules, rich in antioxidants, and possessing a well-balanced lipid profile [24]. In this regard, incorporating olive oil coproducts into farm animal diets has improved animal welfare, shelf life, and quality of animal products attributed to their polyunsaturated fatty acid (PUFA) and polyphenol content [25,26,27,28].

Several studies highlighted the benefits of incorporating long-chain n-3 PUFAs into the diet, establishing an optimal n-6/n-3 PUFA ratio of 3:1/4:1 to improve the health status not only of animals but also of humans as end consumers [13]. Indeed, it has been shown that fatty acid content and composition are directly influenced by dietary intake [29,30]. Considering this, Ponnampalam et al. [29] emphasize that diet plays a pivotal role in lipid metabolism and fatty acid (FA) synthesis in both animals and humans, and the effects of fatty acids are determined by their energy concentration and type. In addition, pasture- and forage-based diets are rich in omega-3 (n-3) short-chain PUFAs, while concentrated feed diets are common sources of omega-6 (n-6) short-chain PUFAs. The presence of n-3 and n-6 PUFAs in animals results from their direct consumption (feed) or the synthesis of longer-chain PUFAs from short-chain precursors in the body through desaturation and elongation processes. The consumption of n-3 PUFAs is known to improve the health and welfare status of animals and humans due to their various biological, biochemical, pathological, and pharmacological effects. In contrast, high levels of n-6 PUFA consumption can be potentially harmful [29].

Due to the numerous reported beneficial effects, the use of a local resource with high functional value, such as *Olea europaea* extract included in the pigs’ feed ration, could ensure greater environmental, human, and animal sustainability. Based on these considerations, we hypothesize that supplementing pig diets with molecules derived from the olive oil supply chain can improve the fatty acid profile of pigs without altering their growth performance. The objective of our study was to evaluate the effects of olive leaf extract supplementation in the diet, comparing the results with a standard diet that already includes Sulla as a fodder component.

## 2. Materials and Methods

Animal procedures were reviewed and approved by the Ethical Animal Care and Use Committee of the University of Naples “Federico II” (Protocol No. 99607-2017).

### 2.1. Plant Samples

Olive leaves of the *Ortice* cultivar were collected in March 2024 in an olive grove located in the Valley of the Middle Volturno in the Campania Region (Italy) (41°12′46″ N, 14°24′07″ E; 110 m above sea level in Ruviano (CE)). Leaves were collected and were air-dried according to the procedure reported in Di Meo et al. [10]. Sulla (S, *Hedysarum coronarium*) plants in the vegetative state were collected from the “Carmine Campone” farm located in Castelpoto (Benevento, Italy). The plant samples were identified by Prof. Antonino Pollio of the University of Naples (Table 1).

### 2.2. Animals and Experimental Design

The study was conducted at the “Carmine Campone” farm located in Castelpoto (Benevento, Italy) (41°06′28′′ N, 14°40′18′′ E; 285 m above sea level) on thirty commercial hybrid pigs (*Large White* × (*Landrace* × *Duroc*)) in the finishing period. Male pigs were reared in an intensive system in large adjacent pens and were randomly assigned to two dietary treatments (*n* = 15 per treatment): 15 pigs fed a standard diet (control diet) with Sulla plant (500 g/head/day) (C) and 15 pigs fed the diet (C) enriched with *O. europaea* L. leaf extract (300 mg/head/day) (OL). Each dietary treatment consisted of 3 replicates with five pigs each. All pigs were fed twice daily (7.30 a.m. and 4.30 p.m.), with a total daily ration of 3 kg per animal per day, and water was provided ad libitum throughout the study, with daily monitoring and continuous access to fresh water. The study lasted for 90 days. The pigs were weighed (169 ± 7.90 kg initial body weight, 10 ± 1 months of age) and individually identified. Pigs were weighed every 15 days from the beginning to the end of the experimental trial to determine the average daily gain (ADG) and were slaughtered on the same day in a commercial abattoir, according to the European Union welfare guidelines (Council Regulation (EC) No. 1099/2009). Animals were electrically stunned and exsanguinated. The muscle sampling was carried out within the slaughterhouse, immediately after the animal was stunned and slaughtered. The sampling procedure was performed within the room adjacent to the slaughter line, with three samples taken for each muscle. The carcass weights was recorded, and 24 h after slaughter, the *Longissimus dorsi*, diaphragm, *Semimembranosus*, and *Psoas major* muscles were removed from each carcass. Immediately, the samples were transported to the laboratory and refrigerated for further analysis. The storage was carried out under controlled conditions, at low refrigeration temperatures, to preserve its characteristics until the time of analysis.

### 2.3. Pig Performance Assays

The pigs’ weight and total dry matter intake were recorded for 90 days. Data on daily weight gain and feed conversion ratio were collected for the whole experimental period. Weight was measured prior to the start of the experiment, every 15 days during the experiment, and at the end of the experiment. using a cage with a scale. Feed intake was recorded using a trolley equipped with scales, which enabled the amount of feed supplied to each pen to be recorded. When the pigs were weighed individually, the remaining feed was also weighed to allow an estimation of feed intake. The feed conversion ratio was estimated based on the average daily weight gain and mean daily feed intake per pen (i.e., total pen feed intake/5 pigs per pen).

### 2.4. Standard Diet and Muscle Chemical Composition Analysis

Pigs were fed a standard diet containing maize, soya protein, wheat, barley, bran, and Sulla (*H. coronarium* L.) pasture grasses, available for pasture-fed pigs. The diets were formulated to be isonitrogenous and isoenergetic. The individual feedstuff from each animal category was sampled weekly and analyzed according to the AOAC method [31]. The composition and amounts of the standard diet are shown in Table 2. The standard diet offered to the pigs of the OL group had the same ingredients as the diet supplied to the control group pigs (Table 2).

In addition, the muscle samples were analyzed for moisture, ash, crude protein, and crude fat according to the AOAC methods [31].

### 2.5. Functional Diet: Preparation and Analysis

Dietary supplementation was carried out by adding dried olive leaves of the *Ortice* cultivar, in powder form, in the form of flour, into the standard feed ration. The functional feed was administered to the pigs in the OL group at a concentration of 300 mg/head/day. In this study, the added dose of *O. europaea* extract was selected based on prior research emphasizing its antioxidant properties [10]. Diet and dried olive leaf extracts were obtained by microwave-assisted extraction (MAE), following the method reported by Di Meo et al. [10,25,26]. In the dried leaves, standard diet, enriched diet, and Sulla pasture grasses, the total polyphenol (TPC) and flavonoid (TFC) content and the antioxidant activity were determined as reported in Di Meo et al. [10]; additionally, phenolic compounds in the olive leaves and Sulla extracts were characterized by using HPLC-UV.

### 2.6. HPLC-UV Analysis

#### 2.6.1. Chemicals

The authentic phenolic compounds vanillic acid, gallic acid, tyrosol, 3-hydroxytyrosol, *p*-coumaric acid, caffeic acid, ferulic acid, ellagic acid, oleuropein, chlorogenic acid, kaempferol, quercitin, rutin, verbascoside, luteolin-7-glucoside, and oleuropein diglucoside were purchased from Merck KGaA (Darmstadt, Germania). HPLC was used to check the purity of these compounds, and one peak was given by each compound. All used solvents were of HPLC grade.

#### 2.6.2. Preparation of Extracts with Different Solvents

Leaves of *Ortice* olive tree (25.0 g) and *H. coronarium* plants (25.0 g) were frozen at −80 °C, powdered in a mortar, and left to infuse for three days in different solvents to obtain the extract with hexane (H), methylene chloride (MC), acetone (A), methanol (M), and water (W) [32]. The organic extracts were filtered on Whatman paper n° 1, dried with Na_2_SO_4_, and concentrated under vacuum, yielding 0.4, 1.3, 1.0, and 2.2 g of residual material, respectively; instead, an aliquot (25 mL) of the aqueous fraction was lyophilized to obtain 0.3 g (2.4 g in total) of crude extract (Figure 1).

The organic fractions A and M and the aqueous residue W obtained from olive tree Ortice leaves and from *H. coronarium* plants were injected into HPLC, obtaining the chromatographic profiles to be compared with the standards considered. The methanolic extracts (O-M and S-M) were the most interesting in terms of abundance and number of compounds identified by comparison with the reference standards, which are illustrated in Figure 2.

#### 2.6.3. General Experimental Procedures

The high-performance liquid chromatography (HPLC) apparatus consisted of a System Gold 127 Beckman pump, a System Gold 166 UV detector Beckman, and a Shimadzu Chromatopac C-R6A recorder. Analytical HPLC was performed using a Phenomenex column InertClone 5 um ODS (2) 150 A (150 × 4.60 mm) and eluted with HCOOH (A, 0.1% in H_2_O) and acetonitrile (B), starting with 5% B for 5 min and followed by the installation of a gradient to obtain 100% B over 30 min, at a solvent flow rate of 1 mL/min. Analytical thin layer chromatography (TLC) was performed on Kieselgel 60 F_254_ or RP-18 F_254_ plates with 0.2 mm layer thickness (Merck). Spots were visualized by UV light or by spraying with H_2_SO_4_/AcOH/H_2_O (5:10:4). The plates were then heated for 3 min at 120 °C. Preparative TLC was performed on Kieselgel 60 F_254_ plates, with 0.5 or 1 mm film thickness (Merck). Flash column chromatography (FCC) was performed on Kieselgel 60, 230–400 mesh (Merck), at medium pressure. Column chromatography (CC) was performed on Kieselgel 60, 70–240 mesh (Merck).

### 2.7. Fatty Acids Analysis of Meat and Diet

Different types of pig muscles (*Longissimus dorsi*, Diaphragm, *Semimembranosus,* and *Psoas major*) were selected. In detail, 3 samples for each type of muscle were analyzed for each pig. A cold extraction protocol [25] was used for lipid extraction from food matrices. The homogenized and finely chopped sample was weighed, added to CH_2_Cl_2_, sonicated for 10 min, and placed under agitation for 2 days. So, the sample was filtered, dried over sodium sulfate, and subsequently evaporated to dryness under a nitrogen flow. Feed was extracted with methylene chloride, and the organic phase was dried under a slight flow of nitrogen. The sample was dried for 1 h over P_2_O_5_ and then treated for 20 min at 60 °C with a solution of boron trifluoride/methanol 10% (1.3 M, 0.5 mL) and 100 µL of dimethoxypropane [33]. Finally, each solution was extracted twice with hexane, and the organic phase was dried. The fatty acid methyl esters were redissolved in hexane, filtered on a millet, and injected into a gas chromatograph [25,26].

The gas chromatographic analysis of the fatty acid profile involved the conversion of fatty acids into methyl esters (FAMEs). The transmethylation reaction in the presence of anhydrous methanol could occur with either basic or acidic catalysis. For basic transmethylation, an aliquot of the sample accurately weighed was trans-esterified using the following protocol. The extracted fat was dissolved in *n*-hexane (containing internal standards) and then treated first with a 2N solution of KOH in methanol and then vortexed for 1 min at room temperature. The reaction was stopped after 5 min by adding NaHSO_4_·H_2_O. After phase separation, the supernatant (hydrocarbon phase), containing the methyl esters of fatty acids, was transferred into a vial and then injected into the gas chromatograph. The analysis was performed in triplicate. For acid transmethylation, an aliquot of methylating reagent (acetyl chloride slowly added to anhydrous methanol) containing internal standards was added to the previously separated lipid fraction. The transmethylation reaction took place at 60 °C for 1 h with the reaction vials being agitated at 5 min intervals. After cooling, 2 mL of saturated aqueous solution in hexane and 5 mL of hexane were added and agitated for 1 min, and the hexane phase was recovered. The hexane phase was washed twice with 5 mL of saturated aqueous solution in hexane. The recovered organic phases were dried with 0.5 g of anhydrous sodium sulfate for 1 h. The hexane phase (supernatant) was transferred into a vial and injected into the gas chromatograph. The gas chromatography (GC) apparatus consisted of an Agilent GC 8890A instrument equipped with a split/splitless inlet and FID detector and an Agilent 7693A automatic liquid sampler. Analysis was performed using a capillary column Agilent HP-88 (100 m × 0.25 mm, 0.20 µm film thickness). The following parameters were set during the experiments: injected sample volume, 1 mL, introduced into injector using autosampler heated to 250 °C, with a split ratio of 100:1; nitrogenous as the carrier gas, 40 psi, constant pressure mode; oven ramp program, 100 °C (13 min), 10 °C/min to 180 °C (6 min), 1 °C/min to 200 °C (20 min), 4 °C/min to 230 °C (7 min); detector temperature, 280 °C; H_2_, 40 mL/min; air, 400 mL/min; and make-up gas, 25 mL/min. A mixture standard of 37 FAMEs was diluted to 50–100 ng/µL for each component and used to test the system repeatability. All components were well resolved. The overlaid chromatograms from six injections showed excellent area and RT repeatability. The fatty acids were identified by comparing the retention times with those of reference standards and quantified relative to the internal standards.

### 2.8. Statistical Analysis

Statistical analysis was carried out using GraphPad Prism (GraphPad, San Diego, CA, USA) version 8.0 for Windows. Before statistical analysis, the normality of variable distributions was assessed, and Levene’s test was applied to evaluate the homogeneity of variances. Data on pig meat chemical composition and animal performance were analyzed using a one-way ANOVA (Newman–Keuls post-test) to evaluate the effect of the dietary treatment. Data on the functional analysis of the diet were analyzed using a one-way ANOVA. To compare the fatty acids of meat, as well as the total content of saturated, monounsaturated, and polyunsaturated fatty acids, a two-way ANOVA was performed (followed by a Tukey test). The data were expressed as mean ± SEM, and a *p*-value <0.05 was considered the minimum statistical significance.

## 3. Results

### 3.1. Pig Performances and Chemical Composition of Muscle

Table 3 shows the results obtained from the in vivo performance for both control and treated pigs. Statistical analysis revealed no significant differences between dietary treatments for the analyzed parameters, including the initial and final body weight (at the beginning and end of the experiment), carcass weight, dry matter intake (DMI), average daily gain (ADG), and feed-conversion ratio (FCR). Furthermore, no significant differences were observed in the chemical composition of the different muscles between the two groups (C and OL). The average values for moisture, raw protein, ether extract, and ash content are reported in Table 3, with no significant differences found (*p* > 0.05) across these parameters.

### 3.2. Functional Diet Analysis

#### 3.2.1. Total Polyphenols Content, Total Flavonoid Content, and Antioxidant Activity

The results of the total polyphenols and flavonoid content were quantified as mg gallic acid equivalent/g dry weight (mg GAE/g DW) and mg quercetin equivalent/g dry weight (mg QE/g DW), respectively. Antioxidant activity was assessed based on the percentage inhibition of free radicals. Table 4 details the total phenolic content and antioxidant activity of diet, Sulla, and olive leaf extract. The polyphenols and flavonoids content of the enriched diet (33.59 ± 0.88 mg GAE/g; 2.98 ± 0.32 mg QE/g) increased statistically significantly compared with the standard diet (8.14 ± 0.30 mg GAE/g; 1.71 ± 0.20 mg QE/g) (*p* < 0.001) and Sulla extract (17.47 ± 0.52 mg GAE/g; 3.02 ± 0.26 mg QE/g) (*p* < 0.001). Additionally, the antioxidant capacity of olive leaf extract (51.10 ± 2.98%), enriched diet (standard diet + *Ortice* olive leaf extract; 70.20 ± 2.36%), and Sulla (63.20 ± 2.00%) showed a statistically significant increase compared with the standard diet (28.21 ± 1.20%). These findings highlight the substantial functional benefits of the functional diet with *O. europaea* L. polyphenols due to its high polyphenols content and enhanced radical scavenging activity. Additionally, the data suggested that pasture grasses, such as Sulla, available to the pigs, contributed significantly to the diet’s phenolic compound content, underscoring their value as a natural resource.

#### 3.2.2. Phenolic Composition

The main compounds identified in the phenolic extracts (*Ortice* olive leaves and Sulla plant) are shown in Figure 2. The chromatographic profiles of the methanolic extracts of *Ortice* leaves (Figure 2A) highlighted the presence of several key phenolic compounds, including tyrosol (3), its 3-hydroxy derivative (4), p-coumaric acid (5), oleuropein (9) and its di-glucoside (16), chlorogenic acid (10), and two flavonoids: quercetin (12) and luteolin-7-glucoside (15). The phenylpropanoid verbascoside (14) was also identified [34,35,36]. Notably, Ortice olive leaf extract exhibited the highest relative concentrations of verbascoside (14), luteolin-7-glucoside (15), and oleuropein diglucoside (16), while Sulla was particularly rich in caffeic acid (6) and ellagic acid (8). The methanolic extract of the Sulla plant predominantly contained tyrosol (3), its 3-hydroxy derivative (4), caffeic acid (6), ellagic acid (8), and oleuropein (9) (Figure 2B). The relative concentrations of these phenolic compounds were determined by comparing retention times and absorption spectra with those of pure standards, as shown in Table 5.

The chromatographic analysis highlighted the qualitative molecular difference between *Ortice* olive leaf extract and Sulla, emphasizing the distinctive phenolic profiles of each. These findings underscored the potential benefits of supplementing the pig’s diet with *O. europaea* extracts, particularly when combined with forage sources like Sulla. The data suggested that the bioactive molecules of Sulla associated with the phenolic compounds from the olive leaf extract significantly contributed to the enhancement of the diet’s antioxidant properties.

### 3.3. Fatty Acid Profile

#### 3.3.1. Dietary Fatty Acid Profile

Table 6 shows the content of individual fatty acids in both the standard and enriched diet. It was evident that the diet enriched with *Olea europaea* L. polyphenols had a higher intake of oleic acid (MUFA) and alpha-linolenic acid (PUFA n-3), along with lower levels of saturated fatty acids (SFA). In contrast, the standard diet contained higher levels of saturated fatty acids, particularly palmitic acid, and PUFA n-6.

#### 3.3.2. Muscle Fatty Acid Profile

The effects of the dietary treatment on individual and total fatty acids in pig muscles are shown in Table 7. In the present study, the supplementation of olive leaf extract resulted in a statistically significant reduction in SFA (*p* < 0.001) and an increase in MUFA (*p* < 0.001) across all muscle types. Furthermore, no significant differences were found in the fatty acid composition between the different muscles. The functional diet with extracts of *Olea europaea* L. contributed to a significant reduction in SFA levels, particularly stearic (C18:0; *p* < 0.001) and palmitic acid (C16:0; *p* < 0.001). In the MUFA class, this supplementation significantly increased the levels of oleic acid (C18:1n9; *p* < 0.001), while in the PUFA class, it significantly reduced the levels of linoleic acid (C18:2n6; *p* < 0.001) and α-linolenic acid (C18:3n3; *p* < 0.001). As a result, the MUFA/SFA ratio increased significantly (*p* < 0.001) with OL treatment compared with the control. In addition, the OL diet significantly reduced the total n-6 PUFA content (*p* < 0.01) and did not significantly alter the total n-3 PUFA content in all analyzed muscle types. Finally, the OL diet helped to maintain the n-6/n-3 ratio in the optimal range (*p* < 0.01).

## 4. Discussion

The results of this study suggested that the supplementation of natural antioxidants from olive leaves had a positive impact on the lipid profile of pig meat. In particular, the bioactive molecules coming from olive leaf extracts in the pig feed ration contributed to guaranteeing suitable production performance and improving the meat fatty acid profile with the correct balancing of SFA, MUFA, and PUFA, without apparent negative effects on the animals’ performance.

In this study, the use of dried olive leaves from the *Ortice* cultivar—native to the study region—was selected due to their high concentration of bioactive compounds and potent antioxidant activity [10]. Considering the high chemical stability of dry leaf powder, this extract was mixed with the diet to prevent the animals from rejecting olive leaves [13]. The decision to supplement the pigs’ diet with *O. europaea* molecules, in addition to molecules present in the pasture grasses (Sulla), resulted from a functional and nutritional screening of the compounds (Table 2, Table 4 and Table 5) and the analysis of the lipid profile of the extracts. The results regarding the composition of the selected extract appeared to confirm an appreciable amount of crude protein (25%), ash (11.36%), NDF (36%), and ADF (20.8%), in accordance to numerous bibliographic studies [37,38,39]. As reported by Kadi et al. [38], Sulla, due to its high protein and fiber content, represents a suitable dietary source for animals, with a composition similar to alfalfa; this contributes to ensuring a better overall health status in pigs. Pereira et al. [40] also identified oleuropein and lueolin-7-O-glucoside as the most abundant phenolic compounds in olive leaf extracts. These phenolic compounds are known to impart several beneficial effects in animals of zootechnical interest, including antioxidant, anti-inflammatory, antiproliferative, antibacterial, and antifungal properties [3,26,41]. Furthermore, Sulla, a fodder usually fed to ruminants [18,20], was found to be particularly appealing to the pigs in our study; it has a high content of condensed tannins with an anti-parasitic effect [19], and it also contains phenolic compounds with significant antioxidant capacity, including quercetin, kaempferol, myricetin, and the isoflavone formononetin [16]. In addition, molecular characterization of the Sulla extract, consistent with findings by Molinu et al. [42], revealed the presence of phenolic acids such as caffeic and ellagic acid, known for their beneficial effects in zootechnical animals [43,44,45,46]. Wang et al. [44] point out that a diet rich in ellagic acid enhances antioxidant capacity, digestive enzyme activity, immune function, intestinal function, and growth performance in broilers. Similarly, caffeic acid has been shown to improve intestinal barrier function, modulate gut microbiota and its metabolites, reduce inflammatory responses, and decrease oxidative stress, thereby enhancing growth performance in piglets [45]. Despite these benefits, there is a notable gap in the scientific literature regarding the antioxidant capacity and individual phenolic compound quantification of the Sulla extract. Based on these, we aimed to better understand the molecular profile of this short-lived perennial legume, native to southern Italy, and assess its potential benefits for pigs. Several studies report that the antioxidant effects are known to be influenced by the concentration and type of polyphenols integrated into the diet [5,24,47,48]. Our study confirmed that the polyphenol content in *O. europaea* L. extracts was consistent with previous reports [10,49]. Variations in phenolic composition among olive leaf extracts could be attributed to differences in the analytical extraction procedures and/or sample origin or harvesting period [10,40,50,51]. Specifically, the polyphenol content of the *Ortice* cultivar in our study (19.13 mg GAE/g) aligned with earlier findings indicating that the content for this cultivar averaged around 16 mg GAE/g [10]. Furthermore, our findings showed that supplementation with olive leaf extract provided no change in the growth performance of the pigs, as reported even by Leskovec et al. [52]. Similarly, Botsoglou et al. [50] reported that dietary supplementation with fish oil and olive leaves did not adversely affect pig growth rates, contrary to the results of Paiva-Martins et al. [28], who observed different outcomes with olive leaf supplementation. One of the most expected results of incorporating olive leaf into pig diets was its effect on the fatty acid profile of meat. In our study, positive effects were observed in the fatty acid composition showing a reduction in SFA, an increase in MUFA, and a positive balance of n-6 PUFAs and n-3 PUFAs. Specifically, in our study, the OL diet led to a reduction in n-6 PUFAs and maintained n-3 PUFA levels. Wood et al. [53] report that PUFAs, which cannot be synthesized, exhibit tissue concentrations that respond rapidly to dietary changes; in contrast, major SFAs and MUFAs, being synthesized endogenously, show less sensitivity to dietary variations. However, our study observed that increased dietary inclusion of olive leaf extract promoted a significant reduction in SFA content and a linear increase in MUFA levels, in agreement with the findings of Joven et al. [54]. Joven et al. [54] also reported that diets enriched with olive cakes result in higher MUFA and lower PUFA concentrations in pork, consistent with our results. The increase in MUFA in pigs’ meat, mainly due to the high content of C18:1n9 in olive leaf extract (36.18 g/100 g), supports previous research indicating that dietary oleic acid can influence the quality of pork by improving its sensory characteristics and oxidative stability [13,50]. Numerous studies have reported effects in pigs [55], in broilers [56], and in rabbits [57]. Losacco et al. [57] evaluated the effects of the use of olive by-products as a dietary supplement for rabbit production and health highlighting an increase in MUFA in meat, especially oleic acid, showing that the contents of intramuscular oleic acid and MUFA of rabbits fed different olive pomaces were proportional to the oleic acid content of the by-product. These effects are caused by the presence of oleic acid in the diet as it is known that fatty acid deposition in pigs is mainly influenced by the lipid composition of the diet [58]. In the present study, the overall PUFA content in meat was largely unaffected by dietary treatment, except for the levels of C18:2n6 and C18:3n3, where significant variations were shown. The literature indicates that diets high in PUFAs can enhance meat’s susceptibility to oxidation [53]. This observation underscores the advantage of using *O. europaea* molecules, which are low in PUFAs, as they positively influence the lipid profile of the meat. However, Sarmiento-García et al. [59] emphasize that fatty acid content and composition could vary depending on multiple factors, such as the type of muscle analyzed, genetic variation (with important differences between commercial and indigenous pigs), age, or slaughter weight. Pugliese et al. [60] found that the age of pigs can affect the deposition of MUFA, with indigenous breeds showing an increased capacity for MUFA accumulation. Supplementing pig diets with moderate levels of olive leaves, even at low dosages, is a beneficial source of biologically active compounds and could increase the phenolic content in meat [3]. In addition, it may also positively affect pig performance and fatty acid composition by decreasing the SFA content and increasing the MUFA, especially the proportion of oleic acid, which is also essential for human health [54].

This study contributes to advancing knowledge on the potential mechanisms by which functional feed containing molecules from *Olea europaea* L., administered to pigs during the finishing phase, influences the fatty acid profile of pig meat. Specifically, this pilot study offers promising insights into the use of functional feed enriched with bioactive compounds from *Olea europaea* L. to modify the fatty acid composition of pig meat. As highlighted by several scientific studies, various physiological, biochemical, and nutritional factors influence these parameters, including the modulation of lipid composition; reduction in lipid oxidation; regulation of enzymes involved in digestion, metabolism, and lipid synthesis; modulation of the inflammatory response; and hormonal regulation of insulin, cortisol, and thyroid hormones [13,23]. The combined effects of these factors could contribute to the production of meat with an improved lipid profile, as well as enhanced growth efficiency, performance, and overall animal welfare.

## 5. Conclusions and Future Implications

This study is the first to focus on the inclusion of a functional feeding approach in finishing pigs. Our research highlights that (i) Sulla extract and olive leaf extract have a high antioxidant capacity and high polyphenol content, (ii) the functional diet had no significant effects on the growth performance of pigs and the chemical composition of meat, and (iii) the functional diet improved the fatty acid profile of pigs’ meat.

The introduction of sustainable pig feeding models could contribute to the ecological transition of territories with large olive and cereal livestock production in inland Mediterranean areas, offering further insights into the application of strategies for the recovery and use of beneficial molecules for the formulation of functional feeds in terms of resource circularity in olive-growing areas.

This study would represent a real strategic approach to the recovery of an agro-industrial residue in line with the circular economy concept to stimulate the feed industry to produce functional feeds for large-scale distribution.

Based on these premises, the scientific community could start research aimed at investigating the effects of these molecules on the functional quality, rheological characteristics, and oxidative stability of meat, as well as the morpho-functional aspects of various organs and tissues involved in animal welfare. Further research is needed to confirm the results obtained in this study and to extend knowledge about the effects of feed functionalized with natural bioactive compounds.

## Figures and Tables

**Figure 1 animals-15-00535-f001:**
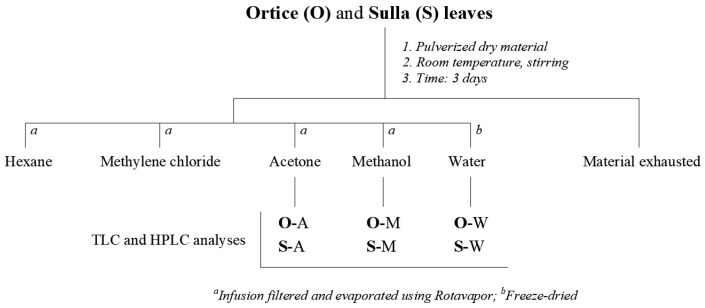
Extraction of plant material.

**Figure 2 animals-15-00535-f002:**
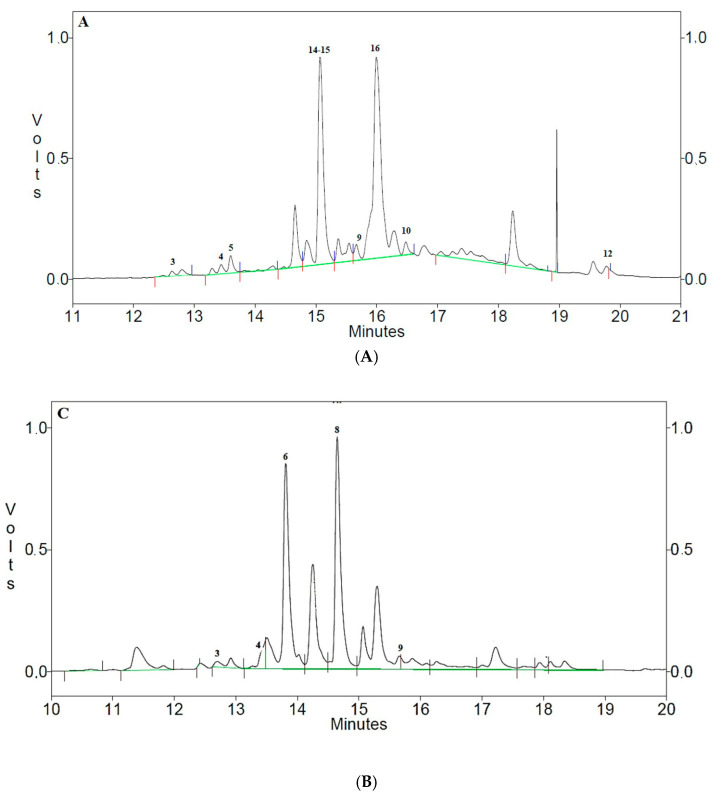
Chromatographic profiles of methanol extracts of *Ortice* olive leaves (**A**) and Sulla plants (**B**).

**Table 1 animals-15-00535-t001:** Plants under study.

Family Name	Cultivar Name	Scientific Name	English Name
Oleaceae	Ortice	*Olea europaea*	Olive tree
Brassicaceae	-	*Hedysarum coronarium*	Wild radish

**Table 2 animals-15-00535-t002:** Ingredients and chemical composition of standard diet and pasture grasses on dry matter (% D.M.).

*Standard Diet*	
Components	% D.M.
Maize	45
Barley	27
Wheat	15
Soy protein 46%	8
Bran	5
**Parameters**	**% D.M.**
Moisture	11.15
Dry matter	88.85
Crude protein	13
Ash	1.94
Crude fats	2.78
Raw fiber	4.05
Starch	52.17
** *Pasture grasses* **	
**Components**	**% D.M.**
Sulla (*H. coronarium* L.) ^a^	18.24
**Parameters**	**% D.M.**
Moisture	81.76
Dry matter	18.24
Crude protein	25
Ash	11.36
NDF ^1^	36
ADF ^2^	20.8

^a^ 500 g/head/day of Sulla (*H. coronarium* L.) pasture grasses; ^1^ NDF, neutral detergent fiber; ^2^ ADF, acid detergent fiber.

**Table 3 animals-15-00535-t003:** Pig performances and chemical composition of muscle *Longissimus dorsi* (g/100 g wet weight).

	Dietary Treatment		SEM ^1^	*p*-Value
	Control	OL		
Initial BW, kg	169	167	1.980	0.194
Final BW ^2^, kg	206	202	2.006	0.161
Carcass weight, kg	199	183	5.160	0.504
Total DMI ^3^, g/d	3.34	3.19	0.125	0.685
ADG ^4^, g/d	754	660	8.203	0.686
FCR ^5^	4.38	4.83	0.129	0.101
**Chemical composition**				
Moisture	71.60	72.00	0.231	0.334
Crude protein	23.98	23.71	0.128	0.090
Ether extract	3.16	3.11	0.101	0.190
Ash	1.52	1.50	0.066	0.156

^1^ SEM, standard error of means; ^2^ BW, body weight; ^3^ DMI, dry matter intake; ^4^ ADG, average daily gain; ^5^ FCR, feed conversion ratio.

**Table 4 animals-15-00535-t004:** Total phenolic content and antioxidant activity (% DPPH inhibition) in standard and enriched diet, *Ortice* olive leaf, and Sulla extracts. ^1^

Phenolic Extracts	TPC (mg GAE/g)	TFC (mg QE/g)	Inhibition %
Standard diet	8.14 ± 0.30	1.71 ± 0.20	28.21 ± 1.20
Olive leaf extract	19.13 ± 0.75 **	3.50 ± 0.43 **	51.10 ± 2.98 **
Enriched diet	33.59 ± 0.88 ***	2.98 ± 0.32 ***	70.20 ± 2.36 ***
Sulla (*H. coronarium*)	17.47 ± 0.52 **	3.02 ± 0.26 **	63.20 ± 2.00 **

^1^ Data are expressed as mean ± SD (*n* = 3). Statistical significance was attributed by one-way ANOVA analysis between the different phenolic extracts ** *p* < 0.01, *** *p* < 0.001).

**Table 5 animals-15-00535-t005:** Phenolic acids contents (mg/g of dry extract) of *Ortice* olive leaves (**A**) and Sulla plant (**B**) with retention time and *r*^2^. Values are mean ± SD (*n* = 3).

**A.** *Ortice* olive leaves					
**No. Peak**	**Compound**	**λ** **(nm)**	**RT (min)**	** *r* ** ** ^2^ **	**Phenolic Compound** **Concentration (mg/g)**
14	Verbascoside	280	14	0.98	0.098 ± 0.05
15	Luteolin-7-glucoside	280	15	0.99	0.098 ± 0.04
16	Oleuropein diglucoside	280	16	0.99	0.100 ± 0.09
**B.** Sulla plant					
**No. Peak**	**Compound**	**λ (nm)**	**RT (min)**	** *r* ** ** ^2^ **	**Phenolic Compound Concentration (mg/g)**
6	Caffeic acid	280	14	0.99	0.068 ± 0.04
8	Ellagic acid	280	14.7	0.99	0.096 ± 0.01

**Table 6 animals-15-00535-t006:** Fatty acid profile of the standard and enriched diet ^1^.

Fatty Acid	Items	Standard Diet (g/100 g)	Enriched Diet (g/100 g)
Myristic	C14:0	0.88	0.70
Palmitic	C16:0	39.44	37.01
*Trans*-palmitoleic	C16:1n7t	0.09	0.05
Palmitoleic	C16:1n7	0.15	0.18
Stearic	C18:0	9.73	11.01
Oleic	C18:1n9	16.35	19.30
*Trans*-linoleic	C18:2n6t	0.30	0.18
Linoleic	C18:2n6	30.96	29.11
*Gamma*-linolenic	C18:3n6	0.16	0.12
Eicosenoic	C20:1n9	0.10	0.08
*Alpha*-linolenic	C18:3n3	0.08	1.01
Eicosadienoic	C20:2n6	0.19	0.08
Dihomo-γ- Linolenic	C20:3n6	0.12	0.10
Arachidonic	C20:4n6	0.77	0.68
Lignoceric	C24:0	0.15	0.14
Eicosapentaenoic (EPA)	C20:5n3	0.23	0.05
Nervonic	C24:1n9	0.06	0.06
Docosahexaenoic (DHA)	C22:6n3	0.01	0.01
*Total*			
∑SFA		50.20	48.86
∑MUFA		16.84	19.67
∑PUFA		32.96	31.47
∑n-6 PUFA		32.50	30.27
∑n-3 PUFA		0.32	1.07
n-6/n-3 PUFA		101.56	28.29
∑SFA/∑MUFA		2.98	2.48
∑SFA/∑PUFA		1.52	1.55

^1^ Abbreviation: MUFA, monounsaturated fatty acid; SFA, saturated fatty acid; PUFA, polyunsaturated fatty acid; n-6, omega6; n-3, omega3.

**Table 7 animals-15-00535-t007:** Fatty acid profile of *Longissimus dorsi*, Diaphragm, *Semimembranosus,* and *Psoas major* muscles ^1^.

Fatty Acid	Control Diet	SEM	*p*-Value	
Myristic	2.13	0.04	0.9956	
Palmitic	27.11	0.28	<0.001	
*Trans*-palmitoleic	0.04	0	0.9988	
Palmitoleic	0.15	0	0.9999	
Stearic	7.79	0.23	<0.001	
*Trans*-oleic	0.01	0	0.9978	
Oleic	51.16	0.59	<0.001	
*Trans*-linoleic	0.03	0	0.9999	
Linoleic	8.02	0.16	<0.001	
*Gamma*-linolenic	0.2	0.01	0.9978	
Eicosenoic	0.16	0	0.9982	
*Alpha*-linolenic	2.13	0.08	<0.001	
Eicosadienoic	0.13	0.01	0.9984	
Dihomo-γ-linolenic	0.06	0	0.9999	
Arachidonic	0.46	0	0.9999	
Lignoceric	0.3	0.01	0.9986	
Eicosapentaenoic (EPA)	0.01	0	0.9999	
Nervonic	0.01	0	0.9999	
Docosatetraenoic	0.07	0	0.9999	
Docosatetraenoic-N6	0.01	0	0.9999	
Docosatetraenoic-N3	0.04	0	0.9999	
Docosahexaenoic (DHA)	0.01	0.01	0.9999	
*Total*				
SFA	37.33	0.54	<0.001	
MUFA	51.53	0.59	<0.001	
PUFA	11.14	0.05	0.8916	
Omega6	8.96	0.14	<0.05	
Omega3	2.18	0.09	0.183	
n6/n3	4.11	0.16	<0.01	
** *Musculus Semimembranosus* **				
**Fatty Acid**	**Control Diet**	**OL Diet**	**SEM**	***p*-Value**
Myristic	2.14	1.57	0.05	0.0829
Palmitic	25.76	21.86	0.36	<0.001
*Trans*-palmitoleic	0.04	0.06	0	0.9999
Palmitoleic	2.24	2.91	0.06	<0.05
Stearic	12.43	11.25	0.11	<0.001
*Trans*-oleic	0.02	0.12	0.01	0.9999
Oleic	36.09	41.27	0.47	<0.001
*Trans*-linoleic	0.18	0.25	0.01	0.9999
Linoleic	12.45	10.06	0.22	<0.001
*Gamma*-linolenic	0.04	0.06	0	0.9999
Eicosenoic	0.01	0.02	0	0.9999
*Alpha*-linolenic	4.04	5.07	0.09	<0.001
Eicosadienoic	0.65	0.95	0.03	0.9516
Dihomo-γ-linolenic	0.01	0.01	0	0.9999
Arachidonic	0.51	1	0.04	0.2522
Lignoceric	2.4	2.48	0.01	0.9999
Eicosapentaenoic (EPA)	0.12	0.12	0	0.9999
Nervonic	0.08	0.14	0.01	0.9999
Docosatetraenoic	0.01	0.01	0	0.9999
Docosatetraenoic-N6	0.45	0.42	0	0.9999
Docosatetraenoic-N3	0.24	0.26	0	0.9999
Docosahexaenoic (DHA)	0.1	0.12	0	0.9999
*Total*				
SFA	42.73	37.16	0.51	<0.001
MUFA	38.48	44.52	0.55	<0.001
PUFA	18.79	18.32	0.04	0.8763
Omega6	14.29	12.75	0.14	<0.01
Omega3	4.5	5.57	0.1	0.0993
n6/n3	3.18	2.29	0.08	0.2527
**Fatty Acid**	**Control Diet**	**OL Diet**	**SEM**	***p*-Value**
Myristic	1.98	1.11	0.08	<0.01
Palmitic	25.01	22.2	0.26	<0.001
*Trans*-palmitoleic	0.03	0.07	0	0.9994
Palmitoleic	3.23	3.65	0.04	0.7396
Stearic	10.24	8.86	0.13	<0.001
*Trans*-oleic	0.01	0.08	0.01	0.9999
Oleic	46.8	53.77	0.64	<0.001
*Trans*-linoleic	0.05	0.14	0.01	0.9996
Linoleic	6.78	5.16	0.15	<0.001
*Gamma*-linolenic	0.1	0.07	0	0.9985
Eicosenoic	0.02	0.06	0	0.9989
*Alpha*-linolenic	2	2.97	0.09	<0.001
Eicosadienoic	0.75	0.31	0.04	0.6699
Dihomo-γ-linolenic	0.13	0.1	0	0.9999
Arachidonic	0.49	0.35	0.01	0.9999
Lignoceric	2.01	0.88	0.1	<0.001
Eicosapentaenoic (EPA)	0.08	0.05	0	0.9999
Nervonic	0.11	0.06	0	0.9999
Docosatetraenoic	0.12	0.05	0.01	0.9999
Docosatetraenoic-N6	0.02	0.01	0	0.9999
Docosatetraenoic-N3	0.01	0.02	0	0.9999
Docosahexaenoic (DHA)	0.01	0.02	0	0.9999
*Total*				
SFA	39.24	33.05	0.57	<0.001
MUFA	50.21	57.69	0.68	<0.001
PUFA	10.55	9.25	0.12	0.0994
Omega 6	8.44	6.19	0.21	<0.001
Omega 3	2.11	3.06	0.09	0.3991
n6/n3	4	2.02	0.18	<0.01

^1.^ Abbreviation: MUFA, monounsaturated fatty acid; SFA, saturated fatty acid; PUFA, polyunsaturated fatty acid; n-6, omega6; n-3, omega3; SEM, standard error of the mean; ns, not significant. Note: Two-way ANOVA in the *Longissimus dorsi*, diaphragm, *Semimembranosus,* and *Psoas major* muscles in control and OL diet.

## Data Availability

All the data are available in the manuscript.

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
