# Peer review of "Functional Feed with Bioactive Plant-Derived Compounds: Effects on Pig Performance, Muscle Fatty Acid Profile, and Meat Quality in Finishing Pigs"

_animals, 2025, doi:10.3390/ani15040535_

Round 1
Reviewer 1 Report (New Reviewer)
Comments and Suggestions for Authors
This study investigated the effects of bioactive plant-derived compounds on the growth performance, muscle fatty acid profile, and meat quality of finishing pigs. The detailed comments are listed as follows:
1. Too many keywords. Keywords should be adjusted 3-5.
2. The Sulla plant and olive leaf extracts used in this study should have more explanation about the ingredients and compositions.
3. Why the basal diet group was not considered for this trial? Without a control group (basal diet), how can the authors conclude their findings about whether the supplements improved meat quality?
4. There is no information on the Sulla plant supplementation dose (lines 160-163).
5. What is the meaning of pigs being fed equal amounts of the same basal diet (line 165)?
6. How feed supplements were provided to pigs? Topping on the diets or other methods? Needs more details.
7. Please clarify why the pigs were fed twice a day. How much feed was supplied per day? Wasn’t it ad libitum access to the feed?
8. The sampling description is too simple. What was the stunning condition? Which locations were selected for muscle sampling? The amounts of sampling? Sample preservation conditions? Need more details.
9. It is confusing that how pigs weights were measure. In the Animal and experimental design (lines 168-171), authors stated that pigs were weighted every 15 days, while in the Pig performance assays indicated that pigs weight was recorded before the start of the experiment and at the end of the experiment (178-180). Authors should revise this issue for better clarity.
10. P values should be incorporated in Table 4.
11. Tables 6 and 7. Actual P values should be included rather than indicating "ns".
12. The discussion should be improved. The authors mostly compared their findings with previous works. However, it is necessary to discuss possible explanations by which the supplements influence their current study findings.
13. The conclusion and future implications should be written more concisely.
Author Response
Please see the attachment.

Reviewer 2 Report (New Reviewer)
Comments and Suggestions for Authors
Overall, this is a well-written manuscript that will help the pork industry gain insights on plant extracts. However, there are some issues.
In the objective, the authors stated formulating a functional feed combining molecules from OL and pasture grass, but this study did not have a negative control. Therefore, we can only tell the difference between with or without OL, not the combination. In many places, for example the hypothesis, the focus is purely on olive leaf, and in the conclusion, the combination was mentioned. So the wording should be changed throughout the manuscript to be consistent.
Line 28: I think only 15 pigs were fed OL.
Line 34: Avoid using words like Excellent because it is very subjective and not suitable for scientific publications.
Line 379-385: I’d suggest listing all main compounds and highlighting the abundant ones.
Line 396-402: I am not sure why are you comparing diets vs individual ingredients (extracts) in Table 6. Diet C vs Diet OL, and OL vs Sulla should be 2 different comparisons. Also, I doubt if you need to conduct statistical analysis for these compounds. Even if there are multiple samples, they are basically from 1 bag. There is no treatment effect, the differences are just lab variations among different sub-samples.
Round 2
Reviewer 1 Report (New Reviewer)
Comments and Suggestions for Authors
The authors have addressed all comments and suggestions. The quality has been improved and can be accepted in its current form.
This manuscript is a resubmission of an earlier submission. The following is a list of the peer review reports and author responses from that submission.
Round 1
Reviewer 1 Report
Comments and Suggestions for Authors
The manuscript is entitled: Functional feed with bioactive plant-derived compounds and their effects on pig performance, chemical composition, and muscle fatty acid profile in finishing pigs. I recommend to the authors a shorter title.
In the simple summary and abstract you did not mention the Sulla plant added in both diets. You should add this information, so is important. Authors did not mention the age and weight of the animals. Moreover, they did not describe the experimental design and number of animals. Please present it in the abstract.
The introduction reports a series of statements, all scientifically correct but unrelated. It does not clearly define the purpose of the experiment, namely the achievement of sustainable pig production. Some information and bibliographical references are repeated in the discussion and no new data seems to be presented in this experiment. The authors should express better the ethical goal achieved through using a local feed extract resource. This part should be shorter and concise.
Material and Methods:
Line 151- Same information in Material and Methods (L 151) and discussion (line 441-442). Delete one.
Lines 160-162-There were 10 pigs, 5 pigs per treatment and three replicates (It is unclear). How are the pens? How many animals per pen? Indoor or outdoor?
Line 172: How were weighed the pigs (machine, no data available)? How you control the feed intake (manual or electronic ear tag? Provide more information.
Table 2 and 3 (a &b) should be mixed and matched.
Figure 2- remove it. Figures not needed.
Line 256- Was the fat test only intramuscular or was there also subcutaneous fat?
Line 258 – “an in-house method” It has a validation? Please, give information in a previous publication reference, as regard validate the method.
Results and discussion
For result and discussion, it is necessary to carry out a more in-depth correlation analysis based on the research results. In my opinion, in order to evaluate the effect of the addition of Sulla and olive extract, it would also have been interesting to evaluate the accumulation of antioxidants in the pork meat.
Lines 316-318- Remove p-value. They are provided in Table 4
Line 321. Tab.4 (why abbreviation here?).
Table 5. Delete “g DMI/g ADG”
Line 388-394 and table 6. There are a format problem.
Table 8, Delete “Items”. Same information as “fatty acid”.
Line 442. Same information in material and methods, delete.
Delete, not necessary: Leaf drying decreases water activity, reduces the enzymatic destruction of certain polyphenols, and increases oleuropein synthesis [12,34].
Line 443-446- Rewrite the long sentence.
Line 449-451- This information should be reported in results.
Line 457-468- This information has not studied in your study.
Line 472-475- Remove it.
Line 477-480- Use this information at the introduction part.
Line 531-536- Authors should reduce their expectations about their experiment.
Conclusions and Future implications.
In general, the conclusions and implications drawn by the authors are not supported by the experiment conducted. They should be more modest and should be considerably more in line with the findings of the present work.
Reviewer 2 Report
Comments and Suggestions for Authors
The authors noted the importance and expediency of studying new phytobiotic feed additives in animal diets in order to enrich animal feed products (pork) with essential fatty acids. In my opinion, this is an actual and valuable addition for agriculture. Because it will help to replace more efficient and cheaper feeds with alternatives from a more profitable and economical side. In general, the work is relevant and of great interest for further development.
However, during the review process, some inaccuracies and recommendations to the authors were noted, to which we would like to receive a full answer:
1. It should be described how the meat productivity of pigs was evaluated.
2. It is not clear whether sulla herb is included in the control diet.
3. I think that Figure 2 is superfluous here.
4. Specify the chemical composition of which part of the body (muscle) is shown in Table 4.
5. In table 4, it is necessary to indicate the live weight at the beginning of the experiment.
6. What is the reason that the use of your supplement is the difference between the final live weight and the carcass weight in the first group, the difference is 5 kg, and in the second group 15 kg?
7. The chapters, in my opinion, should be arranged in the order "feed - pig productivity - meat quality - etc..."
Reviewer 3 Report
Comments and Suggestions for Authors
The aim of this study was to investigate the effects of olive leaf supplementation on the performance and fatty acid profile of pigs. The results showed that it changed the lipid characteristics of meat. Its research significance has certain value and provides a certain basis for its application. However, there are some doubts about the statistical basis of “N = 5”, and the following problems need to be modified:
1. Please modify the layout of the full text.
2. What is the statistical basis for the n = 5 used in the article.
3. In my opinion, the content of the preface lacks logical relationship and is too long, so I suggest that this part should be reorganized and continued.
4. It is suggested to supplement the sex information of Animals in the section "2.2. Animals and experimental design".
5. In order to facilitate readers to read or repeat, please give a brief description of the measurement method of the relevant indicators in 2.3. In addition, the performance of pigs includes not only production performance, but also reproductive performance, etc. Is it possible to explain the performance of pigs only by several indicators in the article?
6. It is recommended to complete the Sulla diet components in Table 2.
7. Supplement specific information such as item number for reagents or instruments appearing in the material section.
8. Which muscle does the chemical composition in Table 4 refer to? Did the author not measure several parts of the sample?
9. Is the representation of P < 0.00 in Table 7 appropriate?
Comments on the Quality of English Languagenone
Round 2
Reviewer 1 Report
Comments and Suggestions for Authors
The study should be considered a preliminary study.
The introduction is too long. It mentions about antibiotics and antioxidant effects, but the results do not include data on these. Eliminate them.
There are no replicates of the experiment. The pen with five animals is considered as one experimental unit. If this is not the case, please clarify.
